Histopathology of crustose coralline algae affected by white band and white patch diseases

Quéré Gaëlle 1 2 queregaelle@gmail.com
Meistertzheim Anne-Leila 2
Steneck Robert S. 3
Nugues Maggy M. 2 4
1 Leibniz Center for Tropical Marine Ecology (ZMT) , Bremen , Germany
2 Laboratoire d’Excellence ‘CORAIL’ and USR 3278 CRIOBE EPHE-CNRS-UPVD , Perpignan Cedex , France
3 Darling Marine Center, School of Marine Sciences, University of Maine , Walpole, ME , USA
4 Carmabi Foundation , Piscaderabaai z/n, Willemstad , Curaçao
Medina Mónica
Electronic publication date: 2015 Jun 30
Publication date: 2015
Volume: 3
Electronic Location ID: e1034
Received 2015 Mar 22; Accepted 2015 May 28
Copyright: © 2015 Quéré et al.
Copyright year: 2015
Copyright holder: Quéré et al.
License: This is an open access article distributed under the terms of the Creative Commons Attribution License, which permits unrestricted use, distribution, reproduction and adaptation in any medium and for any purpose provided that it is properly attributed. For attribution, the original author(s), title, publication source (PeerJ) and either DOI or URL of the article must be cited.
License URL: https://creativecommons.org/licenses/by/4.0/

Keywords: Crustose coralline algae, Disease, Cell death, Boring fauna, Lesion, Histopathology, Regeneration

Funding: European Union 7th Framework programme (P7/2007-2013) CNRS Chaire d’Excellence The research leading to these results has received funding from the European Union 7th Framework programme (P7/2007-2013) under grant agreement No. 244161. MMN also acknowledges support from the CNRS Chaire d’Excellence. The funders had no role in study design, data collection and analysis, decision to publish, or preparation of the manuscript.

==============================
Crustose coralline algae (CCA) are major benthic calcifiers that play crucial roles in marine ecosystems, particularly coral reefs. Over the past two decades, epizootics have been reported for several CCA species on coral reefs worldwide. However, their causes remain often unknown in part because few studies have investigated CCA pathologies at a microscopic scale. We studied the cellular changes associated with two syndromes: Coralline White Band Syndrome (CWBS) and Coralline White Patch Disease (CWPD) from samples collected in Curaçao, southern Caribbean. Healthy-looking tissue of diseased CCA did not differ from healthy tissue of healthy CCA. In diseased tissues of both pathologies, the three characteristic cell layers of CCA revealed cells completely depleted of protoplasmic content, but presenting an intact cell wall. In addition, CWBS showed a transition area between healthy and diseased tissues consisting of cells partially deprived of protoplasmic material, most likely corresponding to the white band characterizing the disease at the macroscopic level. This transition area was absent in CWPD. Regrowth at the lesion boundary were sometimes observed in both syndromes. Tissues of both healthy and diseased CCA were colonised by diverse boring organisms. Fungal infections associated with the diseased cells were not seen. However, other bioeroders were more abundant in diseased vs healthy CCA and in diseased vs healthy-looking tissues of diseased CCA. Although their role in the pathogenesis is unclear, this suggests that disease increases CCA susceptibility to bioerosion. Further investigations using an integrated approach are needed to carry out the complete diagnosis of these diseases.

Introduction

Scientific awareness that marine diseases represent a major threat to coral reefs has led to the multiplication of disease investigations over the past three decades (Weil, 2001; Harvell et al., 2007; Pollock et al., 2011; Burge et al., 2014). Field monitoring surveys have considerably increased our knowledge about macroscopic characteristics, abundance and distribution of coral reef diseases and the environmental factors influencing their dynamics (Gladfelter, 1982; Kuta & Richardson, 1996; Hayes & Goreau, 1998; Nugues, 2002; Willis, Page & Dinsdale, 2004; Aeby et al., 2008; Weil, Croquer & Urreiztieta, 2009; Haapkylä et al., 2010; Tribollet, Aeby & Work, 2011). However, little progress has been made in elucidating disease causation due to the lack of microscopic pathology (Work & Meteyer, 2014). Coupled with microbial culture and molecular essays, histopathology appears as a crucial tool to determine the association between a pathogen and a tissue lesion. It is therefore a vital step in any effective coral reef disease survey (Work & Meteyer, 2014). It provides insight into cell pathology and host response to help resolve the question of disease causation (Work et al., 2014). It can detect etiological microorganisms and propose or refute potential causative agents by their observation in situ. Furthermore, it provides a great amount of information on the cell and tissue damages associated with gross lesions (Peters, 1984; Ainsworth et al., 2007a; Burns & Takabayashi, 2011; Williams et al., 2011; Sudek et al., 2012). Sometimes, even in the absence of pathogens, changes in the host tissue histology hint at the type of infection and lead to a diagnostic (Gupta et al., 2009). It is therefore the only current diagnostic tool that allows the establishment of a link between the potential causative agent and the specific changes in cell and tissue (Work & Meteyer, 2014). For instance, histology has confirmed the association between a fungus and the blue-black band lesion in crustose coralline algae (CCA) affected by the Coralline Fungal disease (CFD) (Williams et al., 2014). However, an integrated approach (i.e., combining microbiological, microsensor, molecular and physiological techniques) is necessary in order to incriminate infectious agents as disease causation and thus complete the diagnostic picture (Richardson et al., 2001; Work & Meteyer, 2014).

Unfortunately, investigations at the cellular level are seriously lacking in diseases affecting CCA despite the importance of these calcifying algae in marine ecosystems, especially coral reefs. Along with scleractinian corals, CCA are important primary producers (Adey & Macintyre, 1973; Chisholm, 2003) and framework builders (Adey & Vassar, 1975) delivering significant functional services in coral reef ecosystems, including enhancing coral larval settlement (Morse et al., 1988; Heyward & Negri, 1999; Harrington et al., 2004; Ritson-Williams et al., 2010; Ritson-Williams et al., 2014). CCA are not spared by the increasing intensity and severity of marine diseases (Littler & Littler, 1995; Hayes & Goreau, 1998) and field investigations on CCA diseases have multiplied in recent years (Aeby et al., 2008; Vargas-Ángel, 2010; Tribollet, Aeby & Work, 2011; Miller et al., 2013; Quéré, Steneck & Nugues, 2015). At present, six disease categories have been reported (Vargas-Ángel, 2010; Williams et al., 2014; Quéré, Steneck & Nugues, 2015), but only CFD and coralline lethal orange disease (CLOD) have known causations. Virtually nothing is known about the other CCA disease categories and they remain histologically uncharacterized. Further knowledge on these diseases and the response of their host could be gained from studies at tissue and cellular levels.

In Curaçao, CCA species are affected by the Coralline White Band Syndrome (CWBS) and the Coralline White Patch Disease (CWPD) (Quéré, Steneck & Nugues, 2015). Both pathologies have the potential to reduce the survivorship and settlement of coral planulae and thus may have important implications for the maintenance and recovery of coral reefs (Quéré & Nugues, 2015). They differ in gross symptoms, spatio-temporal variations and lesion spread, suggesting that they may have different causations (Quéré, Steneck & Nugues, 2015). CWBS lesions are defined by a white-band that appears centrally or peripherally and advances slowly but steadily on the healthy tissue, while CWPD manifests by the presence of distinct white patches on the healthy crust, suggesting sudden losses of tissue (Figs. 1A and 1B). Both diseases result in tissue loss with subsequent colonization by endophytic algae often leading to the death of the diseased patch in the case of CWBS (Quéré, Steneck & Nugues, 2015). Visible symptoms may have a biotic or abiotic origin. On one hand, thermal stress has been shown to cause bleaching in both corals and CCA in the laboratory (Anthony et al., 2008) and algal necroses appear on CCA crust under elevated temperature in aquaria (Martin & Gattuso, 2009). On the other hand, bacterial pathogens can also cause bleaching disease in the marine red algae Delisea pulchra (Fernandes et al., 2011). Gross symptoms in the shape of rings are known to be caused by a bacterial infection in the case of CLOD (Littler & Littler, 1995) and by fungi in the case of CFD (Williams et al., 2014). The aim of this study was to describe CWBS and CWPD at the microscopic level in order to better understand these diseases and their effects on coralline algal tissues.

Figure 1 Gross lesions of CCA diseases.

(A) CWBS in Paragoniolithon solubile and (B) CWPD in Hydrolithon boergesenii from Curaçao in 2012. Black arrow shows the white band in CWBS.

Materials and Methods

Field collection

Crustose coralline algae were sampled in May 2012 at two sites along the leeward coast of Curaçao, Southern Caribbean (12°N, 69°W). Fragments (ca. 10–20 cm2) from four CCA species were collected using hammer and chisel on the reef terrace at 5–10 m depth at two reef sites: Hydrolithon boergesenii, Neogoniolithon mamillare and Paragoniolithon accretum at Water Factory (12°06′32″N, 68°57′14″W) and Paragoniolithon solubile at Playa Kalki (12°22′30″N, 69°09′31″W). Sampling was not targeted towards particular species, but we sought to have an approximately equal number of healthy and diseased samples. A total of 23 fragments, including 7 healthy fragments, 8 fragments affected by CWBS and 8 fragments affected by CWPD, were sampled (Table 1). For each diseased fragment collected, we made sure to incorporate healthy-looking tissue. Each replicate was selected from a distinct patch. Healthy and diseased fragments of each disease were placed in separated collecting bags to avoid contamination and transported in the dark to the laboratory.

Table 1 Number of healthy and diseased fragments collected from each species.

	Healthy	CWBS	CWPD	Total	
Hydrolithon boergesenii	3	1	5	9	
Neogoniolithon mamillare	2	4	3	9	
Paragoniolithon solubile	1	3	0	4	
Paragoniolithon accretum	1	0	0	1	
Total	7	8	8	23	

Histology

Back in the laboratory, a sample (ca. 2–4 cm2) of each fragment was kept for taxonomic identification. The pieces used for taxonomic determination were rinsed with freshwater and dried for six hours in the oven at 60 °C before being checked under a dissecting scope for reproductive and morphological features (Steneck, 1986). The rest was fixed in 4% Formalin-seawater solution and stored in the fridge until further use. Before decalcification, a small piece (ca. 1 cm2) was cut from each fragment so that only the crust of the CCA and a thin (ca. 5 mm) layer of limestone underneath remained. All superficial epibionts (i.e., mostly filamentous algae) present on the surface of the coralline algae were removed. In the case of diseased fragments, each piece was chipped so that it included the boundary between healthy and diseased tissues.

Each sample was then placed in an individual container with 5% L-ascorbic acid solution to gently decalcify over a period up to one week. The solution within each container was refreshed every two days. Once the skeleton and limestone were dissolved, the tissue samples were placed in individual embedding cassettes and dehydrated at room temperature in ascending grades of ethanol (70%, 80%, 95%, 100%, 100%) for 40 min each, followed by an immersion in limonene (three baths of 40 min each). Samples that could not be processed immediately were stored in 70% ethanol for a maximum of 5 days. This additional step did not affect the results (G Quéré, pers. obs., 2014). CCA tissue was then placed in three successive baths of paraffin (Paraplast® Plus™; Sigma-Aldrich, Seelze, Germany) each time 40 min before being embedded into paraffin blocks. Samples were orientated so that transverse sectioning was possible. The blocks were stored overnight at 4 °C in the fridge to ease withdrawal from the cassette the following day. The blocks were sectioned (section thickness 5 and 7 µm) using low profile microtome blades (Leica DB80 LX; Leica Biosystems GmbH, Wetzlar, Germany) mounted on a calibrated rotary microtome (LEICA™ RM2245; Leica Microsystems GmbH, Wetzlar, Germany). Sections were floated onto water (20 °C), mounted onto clean slides and dehydrated on a slide drying bench for minimum 40 min at 50 °C.

Sections were then rehydrated and stained following the Sharman staining series (Sharman, 1943) modified from Ruzin (1999). This method stains the cell walls of plant tissue in tannic acid and iron alum after the protoplasts have been stained in safranin and orange G (see Document S2 for detailed staining procedure). Several other staining methods were tried, but this method was the most effective to visualize the different cellular components of CCA. Sections were then dehydrated in successive baths of ethanol (45%, 90% and 100%) and cleared with limonene. Coverslips were finally mounted using adhesive resin. We examined and photographed 10 permanent histology sections of each CCA fragment using light microscopy (Leica DM750; Leica Microsystems GmbH,Wetzlar, Germany) with integrated camera (Leica ICC50 HD) using the Leica LAS EZ software.

Analyses

Host response was described at the microscopic level and interpreted by comparing normal healthy fragments paired with diseased ones. The presence of invading organisms, their type and localization within the tissue were recorded. In each fragment, organisms could be present in the CCA crust (i.e., epithallus, perithallus and/or hypothallus) or in the limestone underneath the crust. In addition, in diseased fragments, we noted whether they were located in the healthy-looking and/or diseased tissues of the fragments. The identification of the invading organisms was beyond the scope of this study and was restricted to two boring categories: macroborers (i.e., boring sponges, helminths and others) and microborers (i.e., cyanobacteria). Sample sizes were not sufficient to allow robust statistical analysis. All results are reported for pooled species of CCA as the number of replicates per species was too low to make comparisons between species, but species-specific data are listed in Table S1.

Results

The four CCA species presented similar responses towards diseases. For both diseases, we observed no difference in cell structure and organization between healthy tissue of healthy CCA and healthy-looking tissue of diseased CCA. Cell walls and contents in healthy-looking tissue of diseased CCA were intact without any apparent damage (Figs. 2B and 3B). In contrast, the diseased part of the tissue showed distinct histological changes between diseases. Cells affected by both CWBS and CWPD presented no apparent damage of their cell walls, but showed a complete depletion of their protoplasmic content (Figs. 2D and 3D). However, in all cases of CWBS, we observed a transition area between healthy and dead cells consisting of cells that were partially deprived of protoplasmic content (Fig. 2C). Cells containing what appeared to be a condensed nucleus or balled up cytoplasmic materials were also frequently observed (Fig. 4A insert). This transition area most likely corresponds to the white band in the gross morphology (Fig. 1A). It did not exist in CWPD tissue where healthy-looking cells were in immediate vicinity of empty dead cells (Fig. 3C). In two cases of CWBS and one case of CWPD, we observed an overgrowth of the diseased/dead surface by the healthy crust, suggesting tissue recovery (Figs. 4A and 4B).

Figure 2 Transversal histological sections of the CCA, Paragoniolithon solubile affected by CWBS stained in Sharman’s (1943) stain.

(A) Overview with locations of the healthy, boundary and dead areas enlarged in (B), (C) and (D). Note the presence of a transition area with progressive loss of staining from healthy tissue (HT) to dead tissue (DT). B, Boundary; Ep, epithallial cells; Cw, cell wall (silver stain); P, protoplasm (orange to dark stain).

Figure 3 Transversal histological sections of the CCA, Hydrolithon boergesenii affected by CWPD stained in Sharman’s (1943) stain.

(A) Overview with locations of the healthy (HT) and diseased areas (DT) enlarged in (B) and (D). (C) shows the boundary (B) between healthy and diseased areas. Note the absence of a transition area highlighted by sudden loss of staining. B, Boundary; Co, conceptacle; Ep, epithallial cells; Cw, cell wall (silver stain); P, protoplasm (orange to dark stain).

Figure 4 Regrowth of living crust.

Regrowth of living crust in (A) CWBS and (B) CWPD. Remnant healthy crust (red arrows) regrew upward and laterally over dead/dying crust. Insert in (A) displays enlargement of transition area with cells showing a condensed nucleus or protoplasmic content (black arrows). T, Transition; Ep, epithallial cells.

Various macroborers and microborers were observed in both healthy and diseased tissues (Fig. 5). They were more abundant in diseased fragments, particularly in CWBS. Of the 7 healthy fragments examined, 4 (57%) had invading macro- and microorganisms versus all of 8 CWBS fragments and 5 (63%) of the 8 CWPD fragments (Table 2). Of 13 diseased fragments with evidence of boring organisms, sponges were most common (62%) followed by other macroborers (38%) and cyanobacteria (31%). Of the four healthy fragments with borers, 3 had sponges and two had other macroborers and cyanobacteria were not encountered. Within diseased fragments, borers were also more abundant in the diseased tissue of the fragments. Of 13 diseased fragments, 5 (38%) presented borers in their healthy-looking tissue, whereas 12 (92%) showed intrusion by borers in their diseased tissue (Table S1). However, boring organisms were rarely present within or in the immediate vicinity of diseased cells. Boring organisms were more abundant in the underlying limestone than in the CCA crust. In CWPD, borers were found exclusively in the limestone of all 5 diseased fragments containing borers. Cyanobacteria were never seen in the CCA crust. We did not visualize any fungal infections associated with the diseased cells.

Figure 5 Photomicrographs of the most commonly encountered organisms in healthy and diseased CCA.

(A) Boring sponge characterized by silicaceous spicules (red arrow) (B) Unidentified macroborer. Note the CCA cells lining up the burrow suggesting the growth of the algae around the invader (red arrow) and the acellular space around the organism (black arrow). (C) Unidentified macroborer, possibly a juvenile bivalve. (D) Cyanobacterial trichomes (red arrows); (E) Helminth; Cu, cuticule; L, lumen.

Table 2 Number of samples with boring organisms.

Number of samples with boring organisms partitioned by health status of CCA fragments (i.e., healthy, CWBS vs. CWPD), health of tissue within fragment (i.e., healthy vs. diseased) and vertical layer within fragment (i.e., CCA crust vs. limestone).

Health of fragment	Healthy	CWBS	CWPD	
Health of tissue	HT		HT	DT		HT	DT		
Vertical layer	C	L	T	C	L	C	L	T	C	L	C	L	T	
Number of samples			7					8					8	
Samples with borers	1	3	4	2	4	2	6	8		1		5	5	
Samples with sponges		3	3	2	2	3	4	5		1		5	5	
Samples with helminths							1	1						
Samples with other macroborers	1	1	2	2	3	1	3	4						
Samples with cyanobacteria					2		2	3				1	1	
Notes.

HT healthy tissue

DT diseased tissue

C crust

L limestone

T total fragment

Note that the numbers can add up more than for the total fragments since the same fragment may have borers in different sections of the sample.

Discussion

This is the first study providing histological information on CWBS and CWPD. We found no visible difference between healthy tissues of healthy and diseased crusts, which suggests that the action of the disease is localized, at least at the cellular scale. However, variations could occur at a smaller scale. For example, distinct differences in bacterial community between non-diseased corals and healthy-looking tissues of colonies affected by white band disease have been highlighted in Orbicella annularis using molecular techniques (Pantos et al., 2003). In tissue affected by both diseases, the three distinct cell layers characteristic of CCA (epithallus, perithallus and hypothallus) showed cells with an intact cell wall, but depleted from all cytoplasmic content as highlighted by a sudden change in the intensity of the staining. A plausible explanation to cell bleaching is the loss of pigments as already known in corals during bleaching events (Kleppel, Dodge & Reese, 1989). CCA contain phycobilins (phycoerythrin and phycocyanin) pigments that are present in living tissue. Their loss could be followed by tissue necrosis and death (Fernandes et al., 2011). CCA also commonly experience sloughing events (Keats, Knight & Pueschel, 1997). However, the signs detected in this study differ from sloughing. During a sloughing event, epithallial cells are lost or appear loose (Keats, Knight & Pueschel, 1997; Garbary et al., 2013). Here, they remained present in all the diseased fragments as clearly visible in Fig. 3. Additionally, all the different cell layers showed similar changes in the diseased part of the crust (difference in cell staining intensity) whereas in the case of a sloughing event, only the superficial epithallial cells would have shown deterioration.

In CWPD, healthy cells were in immediate vicinity of diseased empty cells whereas in CWBS, a transition area existed where cells had less protoplasmic content than healthy cells as highlighted by a weaker stain within the cells. This transition area could be the sign of a chronic, slowly progressing disease which is reflected in the slow but steady rates of CWBS progression on healthy tissue (i.e., 0.21 ± 0.06 cm month−1 in Quéré, Steneck & Nugues, 2015). In contrast, CWPD generally manifests by a sudden and extensive loss of tissue, often with a rapid turn-over (G Quéré and M Nugues, pers. obs., 2012), characteristic of acute diseases (Work, Russell & Aeby, 2012; McCoy & Kamenos, 2015).

Dead cells were characterized by an intact cell wall and a complete loss of protoplasmic content. In the case of CWBS, some cells in the transition area showed a highly visible nuclei or rounded cytoplasmic content. Histology confirmed cell death but the technique used here did not allow us to determine whether death was the result of necrosis or programmed cell death (PCD). The former is triggered by external factors often affecting many cells within a tissue, while the latter is triggered by intracellular signals activating specific gene expression at the level of a single cell (Greenberg, 1997; Dunn et al., 2012). These phenomena have rarely been studied in multicellular algae (Garbary et al., 2013) and remain poorly understood. Both have been highlighted during bleaching in the sea anemone Aiptasia sp. using a combination of histology, electron microscopy and in-situ end labelling of DNA fragmentation (Dunn et al., 2002). In the transition area, the sudden high visibility of the nuclei or rounded cytoplasmic content could be related to the condensation of the nucleus during PCD. Similar cellular degradation has been observed in Acroporid corals affected by white syndromes (Ainsworth et al., 2007b). However, several other distinct features are necessary to differentiate necrosis (e.g., vacuolization, cell rupture, tissue degradation) and PCD (e.g., cell shrinkage, formation of accumulation bodies) (Dunn et al., 2002; Franklin, Brussaard & Berges, 2006). Interestingly, plants can also present a hypersensitive response that consists of rapid death after infection by a pathogen (e.g., fungi, bacteria, viruses, nematodes) in order to prevent its spread (Garbary et al., 2013). This phenomenon could constitute a plausible explanation for the CWPD symptoms. However, further analyses are required to test this hypothesis.

We observed regrowth of healthy-looking tissue over diseased tissue in both diseases. In reef-building corals, an immune response and repair mechanism consisting of a locally accelerated growth has been shown in wounded colonies (D’Angelo et al., 2012). We could interpret this regrowth as a response of the remaining healthy tissue to counteract the progression of the lesion like a wound healing response in CCA. Similarly, CCA are capable of healing wounds caused by herbivores grazing on their crust by regeneration of perithallial cells within the thallus (Steneck, 1983). This healing response could explain the presence of CCA cells lining up the burrow around the invading organisms (Fig. 5B). CCA may have repaired cells around those damaged by the borer. Alternatively, the algal tissue could have grown around the invaders.

We found various metazoa (sponges, helminth, bivalve juveniles) and microrganisms (cyanobacteria) associated with both healthy and diseased CCA tissue. This is consistent with previous studies which have shown the presence of those organisms in healthy and diseased coral colonies (Work & Aeby, 2011; Séré et al., 2013) and in live and dead coralline thalli (Tribollet & Payri, 2001). These organisms were more abundant in diseased than healthy CCA fragments, and, within diseased fragments, they were more abundant in diseased vs healthy tissue, suggesting a potential link between CCA diseases and the presence of borers. However, it is unknown whether these borers are the cause of the disease or opportunistic secondary colonizers. Among the organisms observed here, several have been identified as pathogenic in other species. This is the case for helminths known to cause tissue loss in Montipora (Jokiel & Townsley, 1974) or cyanobacteria which appear to cause tissue lysis and necrosis in black band diseased corals (Ainsworth et al., 2007a). Ciliates are also frequently associated with diseases and capable of invading animal and plant tissue by breaking cell membranes and walls using enzymes such as proteases (Work & Aeby, 2011). In our observations, boring organisms did not seem to be associated with evident cell pathology, suggesting a secondary invasion. Indeed, diseases may weaken or damage coralline tissues, thus facilitating invasion by borers. The mechanical (chip production) and chemical (dissolution) bioerosion of calcium carbonate by boring sponges or bivalves has been reported (Lazar & Loya, 1991; Zundelevich, Lazar & Ilan, 2007). In our study, the acellular space observed around the different invaders could be due to a digestive effect of the borer on the surrounding CCA cells creating a dead zone around them. It is also possible that the presence of an organism would have weakened the tissue around it leading to its loss during fixation.

Microborers are also well-known agents of bioerosion in live and dead CCA thalli causing higher rates of erosion in dead versus live thalli (Tribollet & Payri, 2001). The same way dead coral skeletons are colonised at the surface and bored inwards, diseased crusts could become rapidly vulnerable to invaders (Tribollet & Payri, 2001). Previous studies looking at the association between host response and potential agents revealed that sponges, cyanobacteria and helminths are absent from acute lesions but often associated with chronic diseases, such as the slowly progressing phases of White Syndromes in Montipora capitata (Work, Russell & Aeby, 2012). Our observations confirm this pattern since sponges were often found spreading through the crust and the limestone in CWBS fragments. In contrast, in CWPD fragments, sponges were exclusively located in the limestone, suggesting that they did not have time to invade the crust. There is evidence that macroborers such as bivalves or sponges could take a couple of years to colonize dead skeleton, as they are long-lived, slow-growing organisms (Tribollet & Golubic, 2011).

The increase of borers within the coralline tissue could have a cascading effect by making carbonate substrata available to new borers, thus increasing their eroding action. Ocean acidification also accelerates reef bioerosion without necessarily affecting the health of boring organisms (Wisshak et al., 2013). Furthermore, synergistic effects of ocean warming, ocean acidification and disease infection enhance the reduction in the calcification rates of CCA (Williams et al., 2014). In the face of climate change, disease outbreaks may thus, together with global stressors and boring organisms, aggravate reef degradation.

Histological observations of lesions from the two diseases did not reveal any evidence for the presence of fungi. A fungus belonging to the subphylum Ustilaginomycetes has been identified as the pathogenic agent responsible for CFD thanks to conventional histology (Williams et al., 2014). We could deduce that fungi are not implicated in CWBS and CWPD. Similarly, fungi were not observed in the white syndrome of Acroporid corals (Ainsworth et al., 2007b). Our method did not allow for the visualization of bacteria which have been identified as causal agents for CLOD (Littler & Littler, 1995). Visualizing bacteria using conventional histology would have required the use of Taylor’s gram stains (Peters, 1983; Work & Rameyer, 2005). Additional techniques such as the embedding of tissue in agar and the use of fluorescence in situ hybridisation (FISH) or transmission electron microscopy (TEM) (Work, Russell & Aeby, 2012) have also been suggested to improve bacterial detectability in coral tissues (Bythell et al., 2002). The same applies for virus-like particules (VLPs) whose presence can be detected using TEM and flow cytometry (Davy et al., 2006). Viruses have been associated with the presence of syncitia inside cells (Work, Russell & Aeby, 2012); however, they were not observed in this study. The potential implication of viruses in coral disease is still unknown but thermally stressed corals produce numerous VLPs (Davy et al., 2006; Rosenberg et al., 2009). Although this study did not identify the agents responsible of the diseases, it allows to narrow the pool of potential suspects. The use of an integrated approach is necessary for further progress on the complete diagnosis of CCA disease.

Conclusions

This study brings a descriptive distinction at the cellular level between CWBS and CWPD. Observations of the diseased tissues were consistent with the signs described in the field. CWBS known to progress slowly but steadily over the CCA in the field showed a transition zone in microscopy. In contrast, CWPD known to cause a sudden loss of tissue in CCA had no transition zone. Although boring organisms were observed at higher abundances within diseased tissues in comparison to healthy ones, we did not find evidence of a direct link between the presence of invaders and the disease lesion. However, the range of potential pathogens could be narrowed as no sign of fungal infection was observed. Standard techniques in histopathology alone cannot elucidate the question of disease causation. Additional methods are necessary to complete the diagnosis picture.

Supplemental Information

Table S1 Type of boring organisms encountered and their localisation within the tissue for each CCA sample

HT, healthy tissue; DT, diseased tissue.

Click here for additional data file.

Supplemental Information 2 Sharman staining procedure (Sharman, 1943) modified from Ruzin (1999)

Click here for additional data file.

We wish to thank the Carmabi foundation and staff for logistic support. Aline Tribollet, Elisabeth Faliex, and Thierry Work kindly assisted with identification of invading organisms and interpretation of the micrographs. We also thank Anna Le Ruz and Dr. Annette Peter for assistance in the laboratory.

Additional Information and Declarations

Competing Interests

Author Contributions

The authors declare there are no competing interests. Gaëlle Quéré is an employee of Leibniz Center for Tropical Marine Ecology (ZMT) and Maggy M. Nugues is an associate scientist of Carmabi Foundation.

Gaëlle Quéré conceived and designed the experiments, performed the experiments, analyzed the data, wrote the paper, prepared figures and/or tables, reviewed drafts of the paper.

Anne-Leila Meistertzheim analyzed the data, contributed reagents/materials/analysis tools, reviewed drafts of the paper.

Robert S. Steneck analyzed the data, reviewed drafts of the paper.

Maggy M. Nugues conceived and designed the experiments, analyzed the data, contributed reagents/materials/analysis tools, wrote the paper, reviewed drafts of the paper.

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
