# Peer review of "Histopathology of crustose coralline algae affected by white band and white patch diseases"

_PeerJ, doi:10.7717/peerj.1034_

## Round 0.1 · original submission · Minor Revisions

· Academic Editor

Minor Revisions

Please address the useful comments provided by both reviewers. In particular, the lack of statistical analysis and the concern about the choice of stain.

Additionally, in order to improve the easiness to read the manuscript, please incorporate the suggestions for grammatical improvement and have an English native speaker check the final version if possible.

Reviewer 1 ·

Basic reporting

• The English in this article was sufficiently clear to allow reader comprehension, but could certainly benefit from further proof-reading to increase clarity and correct minor grammatical errors.
• The article included sufficient introduction and background to demonstrate how the work fits into the broader field of knowledge.
• The structure of the submitted article should conform to an acceptable format of ‘standard sections’.
• Figures were relevant to the content of the article and of sufficient resolution. However, Figure Legends should be revised to increase clarity and to ensure conformity to standard grammatical and stylistic practices (i.e. ensure proper use of capitalization and ensure that references to specific panels are clearly worded).
• The submission is ‘self-contained,’ and represent an appropriate (but somewhat meager) ‘unit of publication’.

Experimental design

• The submission describes the original primary research within the Aims & Scope of the Journal.
• The submission clearly defines the research question, which is relevant and meaningful.
• The investigation was conducted to a reasonable technical standard.
• Methods were described with sufficient information to be reproducible by another investigator.
• The research was conducted in conformity with the prevailing ethical standards in the field.

Validity of the findings

• While this study would have greatly benefitted from larger sample sizes and more diverse analysis (i.e. bacterial community profiling), the data appear robust. However, no statistically analyses were performed. The authors should either perform statistical analyses or explain why none were conducted.
• The conclusions are should appropriately stated and are connected to the original question investigated

Additional comments

Specific comments:
• General: Please proofread with a careful eye towards correcting the multiple grammatical errors present in this manuscript. Below, I have noted 3 specific examples from the first two paragraphs alone:
• Line 28: Change “resolving” to “resolve”
• Line 34: “establishing” to “establishment of”
• Line 50: “causation” to “causations”
• Line 66: You state that his work set out “to complete the diagnostic picture”. However, there is little reason to believe that a single study (particularly once based solely upon histology) could “complete the diagnostic picture”. Please reword.
• Lines 90-91: “All visible epibionts present on the surface of the coralline algae were removed”- Do you have any formation on this community? Macroscopic organisms could also play an important role in this syndrome (e.g. via predation or shading).
• Figures: Letters marking separate panels are hard to see against background. Suggest moving to upper-left to show against white background.
• Figure legends: Indicate the stain/s used. Ensure proper use of capitalization.
• General: Do you have any data on the lesion progression rates in this study?
• Lines 174-188: This paragraph shows a poor understanding of the distinctions between apoptosis, necrosis and cell lysis. Please familiarize yourself with these differences and re-visit this paragraph. You provide evidence to demonstrate that apoptosis is not occurring. However, you also state that necrosis is not occurring, but you do not provide convincing evidence of this. For example, cell lysis (including rupture of cell membranes and dissolution of cell contents) is common in cases of necrosis and the two are far from mutually exclusive.
• Line 196-197: State what these “types of organisms” are.
• 201-207: You suggest that invaders could be either due to “Weakened or damaged coralline [that] may have facilitated invasion by borers” or “Alternatively, although less likely, they could have taken advantage of an existing lesion to invade the CCA.” This a bit confusing as written. The first option suggests that invaders are taking advantage of the disease lesion to invade and the second option appears to suggest exactly the same. Perhaps in the second option you are suggesting that there were lesions present unrelated to the disease? Please clarify.
• Line 214: “However, their presence reveals a weakening of the skeleton.” Is this necessarily true? Could the loss of natural defenses against invaders not provide an alternative explanation for elevated levels of invaders at/near lesion fronts

Reviewer 2 ·

Basic reporting

Missing an official Conclusion section/heading, which is a Standard Section for PeerJ manuscript format. https://peerj.com/about/author-instructions/

Experimental design

The manuscript by Quéré and colleagues addresses important gaps in the area of marine diseases, specifically the characterization of crustose coralline algae (CCA) diseases by histopathology. The gaps the authors highlight are well noted and certainly deserve further research. In total, the methods and findings are descriptive and will help further research on CCA diseases. I would recommend this manuscript to be accepted, given that the authors address the following comments and make the suggested changes.

The method the authors used is a good approach to understanding the physiological effects on the cells, however the authors do not show prior evidence of the particular stains used as appropriate stains for CCA or other Corallinaceae. Please provide rationale or evidence for the reason the particular stains were chosen.

The authors should provide a statement that no obvious differences were noted between species, as clustering them and providing only one image per disease assumes that all samples from varying species had similar cellular responses. If the cellular responses were different, however, those should be reported.

Validity of the findings

The strength of the paper is the histological method, however descriptive comparisons within disease vs. across diseases are hard to follow. The following suggestions may help to clarify the authors’ findings more definitively.
1.) Figures 2 & 3 should be more explicit describing what is being stained.
2.) It would be advisable to present photos from the different samples, so consistent cellular patterns can be observed.
3.) Simplify Table 2. The table is hard to follow and therefore the message this table is trying to convey is being lost.

Table S1 should not be supplemental. It is the most descriptive table the authors present and highlights that while borers are present in diseased samples, they are also present in healthy, therefore borers appear not to be exclusively associated with disease and should be clearly stated.

The authors should answer whether the stain used is appropriate for assessing bacteria in association with CCA. If it is it begs the question, why no bacteria were observed associated with diseased tissue? If it’s not the appropriate stain, then one would expect not to observe bacteria on the histological slides, and sentence on lines 152 & 153 should be removed.

When speculating in the Discussion section, authors should provide multiple possibilities to provide a balanced discussion. For example, further explanation of how the results from Figure 4 conclude that the CCA is growing around the borer should be provided or state an alternative. Perhaps, the CCA grew around the borer or the CCA repaired cells around those that may have been damaged by the borer.

Fixatives and ethanol washes cause tissue cells to reduce turgor. Different materials will react in alternate ways. While a physical cavity could be due to a digestive effect of the borer, it is equally likely that the gap observed between the tissue and the borer could be a result of the fixing process.

How are the signs detected in this study (i.e. difference in cell staining and cellular regrowth) different from the signs after a sloughing event, noted to be quite common for some coralline algae?

Additional comments

The use of histology, a tool proven successful to describe cellular response to disease and health of an organism, is commendable as it is an involved and necessary step to better understanding a disease. The authors provide a descriptive distinction between the two diseases investigated, however the number of samples and the use of histology exclusively may not allow one to address etiology or causative agents of a disease. The distinction of the descriptive methodology paper should be drawn from speculative causative agents.

A conclusion paragraph would be nice to provide a final take away and where the research could assist future studies or gaps that still exist in the discipline.

Additionally, Appendix 1 strongly reflects the protocol's content as well as format outlined at the following link:
http://microscopy.berkeley.edu/Resources/instruction/staining.htm
**Credit should be given to the source and modifications should be highlighted in the Methods section.

Below are minor areas that would benefit the manuscript, if addressed.

Introduction

25-27: Sentence needs editing. In it’s current form it is not clearly understood what the authors are trying to say.
50: Authors should note the 2014 marine disease review by Burge and colleagues.
59: Check Coralline White Band Syndrome acronym (should it be “CWBS”, not CWBD?).

Materials and Methods

96-97: 70°, 80°, etc… should be changed to 70%, 80%...
98: Do the samples stored in 70% ethanol show any differences among your results?

---

## Round 0.2 · accepted · Accept

· Academic Editor

Accept

Congratulations on the acceptance of your manuscript.